# Socazolimab combined with carboplatin and etoposide as first-line treatment for extensive-stage small-cell lung cancer: A cost-effectiveness analysis in China

Lian Tang[1☯], Yong Chen[1☯], Shaoqing Zhan[1], Longxun Zhu[1,2]*, PanFeng Feng [1,2,3,4]*

1 Department of pharmacy, Affiliated hospital 2 of Nantong University, and Nantong First People's Hospital, Nantong, Jiangsu Province, China, 2 Nantong Key Laboratory of Innovative Research on Rheumatology and Immunology, Nantong, Jiangsu Province, China, 3 Nantong Clinical Medical College of Kangda College of Nanjing Medical University, Nanjing, Jiangsu, China, 4 Jiangsu Key Laboratory of New Drug Research and Clinical Pharmacy, Xuzhou Medical University, Xuzhou, China

☯ These authors contributed equally to this work.
* 929083891@qq.com (PFF); 948017729@qq.com (LZ)

## Abstract

### Background

The NCT04878016 trial evaluated the efficacy and safety of socazolimab in combination with carboplatin/etoposide as a first-line treatment for extensive-stage small cell lung cancer (ES-SCLC). This study aims to analyze the cost-effectiveness of this combination regimen from the perspective of the Chinese healthcare system.

### Method

A Markov model with three health states was constructed. The model simulated a time horizon of 10 years with a cycle length of 3 weeks. Costs and utilities were discounted at 5% annually. The primary outcomes were total costs, quality-adjusted life years (QALYs), and the incremental cost-effectiveness ratio (ICER). One-way sensitivity analysis and probabilistic sensitivity analysis were conducted to assess the robustness of the results.

### Results

The base-case analysis showed that the ICER for the socazolimab group compared to the chemotherapy-alone group was 355,316.95 yuan/QALY, which exceeds three times China's per capita GDP in 2024 as the willingness-to-pay (WTP) threshold. One-way sensitivity analysis revealed that PD utility, PFS utility, socazolimab cost, and neutropenia management cost had significant impacts on model results. Probabilistic sensitivity analysis indicated that the probability of socazolimab combined with chemotherapy being cost-effective was 21.9%.

**Data availability statement:** All relevant data are within the paper and its Supporting Information files.

**Funding:** This work was supported by Jiangsu Pharmaceutical Association-Aosaikang Fund (No.A202434), Jiangsu Pharmaceutical Association- Yaoyanxinsheng Fund (No.202564030), Jiangsu Provincial Maternal and Child HealthResearch Institute Fund (No. KYXM (2025) 002), Development Fund of KangDa college of Nanjing medical university (No. KD2024KYJJ294), Scientific Research Project of Nantong Municipal Health and Family Planning Commission (No. MS2024038, QNZ2024026), Jiangsu Key Laboratory of New Drug Research and Clinical Pharmacy fund (No. 25KF03), Nantong Pharmaceutical Association-Yangzijiang Fund (ntyxky2509) and Guangzhou Zhiyi Charity Foundation fund. The funders had no role in study design, data collection and analysis, decision to publish, or preparation of the manuscript.

**Competing interests:** The authors have declared that no competing interests exist.

## Conclusion

At China's WTP threshold, socazolimab combined with chemotherapy is not cost-effective versus chemotherapy alone for ES-SCLC.

## Introduction

Lung cancer stands as one of the malignant tumors with high incidence and mortality rates worldwide [1]. In China, lung cancer ranks first in both the incidence and mortality rates of malignant tumors among men and women. In 2022, there were 1,060,600 new cases of lung cancer, accounting for 22.0% of all malignant tumors, and 733,300 deaths, representing 28.5% of all deaths from malignant tumors [2]. Small cell lung cancer (SCLC) represents about 13% of all lung cancer diagnoses and is recognized as the most aggressive form, characterized by a particularly poor prognosis [3–4]. Patients with this subtype typically have an average 5-year overall survival rate of just 5% [5]. Over 60% of patients diagnosed with small cell lung cancer (SCLC) present with extensive-stage disease at the time of their initial diagnosis [6]. Before the emergence of immunotherapy, the standard first-line treatment for ES-SCLC was chemotherapy based on platinum and etoposide [7]. Etoposide combined with platinum-based drugs is generally used for 4 cycles in first-line treatment, with a maximum of 6 cycles. Longer chemotherapy cycles do not improve survival rates, and there are no better treatment options in second-line treatment either [8]. In recent years, when immune checkpoint inhibitors (ICI) were added to traditional chemotherapy, studies have shown that the overall survival of patients with ES-SCLC was improved [9,10].

Socazolimab exhibits high specificity and affinity for human PD-L1 proteins. By binding to PD-L1, it effectively blocks the PD-1/PD-L1 signaling pathway, thereby disrupting the tumor's immune escape mechanism. This allows T cells to function normally and combat tumor growth [11]. NCT04878016 is a clinical study aimed at evaluating the efficacy and safety of socazolimab in combination with carboplatin/etoposide (EC) chemotherapy in adult patients with histologically confirmed, previously untreated ES-SCLC [8]. The study demonstrated that patients treated with socazolimab exhibited a significant improvement in overall survival (OS) compared to those in the placebo+EC group (13.90 months vs 11.58 months, hazard ratio for death, 0.799; 95% CI, 0.652–0.979; p=0.0158). Additionally, the safety profiles of the two treatment regimens were comparable. The current price of socazolimab is 11,500 yuan (100 mg/4 mL), and it has not yet been included in China's national medical insurance reimbursement directory. To date, there is no definitive conclusion regarding its cost-effectiveness.

Therefore, this study, based on the findings of the NCT04878016 trial, evaluates the cost-effectiveness of socazolimab in combination with chemotherapy as a first-line treatment for ES-SCLC from the perspective of China's healthcare system using cost-utility analysis. The aim is to provide evidence-based recommendations for relevant health policy decisions and clinical practice.

## Method

### Target population and treatment regimen

The inclusion and treatment criteria for the study population are consistent with those of a multicenter, randomized, double-blind, placebo-controlled phase III clinical trial conducted across 54 hospitals in China [8]. Patients will be randomly assigned (1:1) to receive either socazolimab plus EC chemotherapy or placebo plus EC chemotherapy. In both groups, patients will receive intravenous infusions of 5 mg/kg socazolimab or placebo on day 1 of each cycle until disease progression, intolerable toxicity, death, or completion of 2 years of treatment. All patients will also receive four cycles of carboplatin (area under the curve [AUC] 5 mg·min/mL on day 1) and etoposide (100 mg/m$^2$ on days 1, 2, and 3), administered every 3 weeks. The duration of socazolimab or placebo administration is limited to 2 years. Therefore, it is assumed that patients who remain progression-free after 2 years will transition to best supportive care.

### Ethics approval

As this study is entirely based on previous research [8] and publicly available data, it does not include any new research involving human participants or animals by any of the authors, and therefore does not require approval from an independent ethics committee.

### Model structure

A partitioned survival model was constructed using TreeAgePro software (2022) to evaluate the cost-effectiveness of socazolimab combined with EC chemotherapy for ES-SCLC. The model encompasses three health states: progression-free survival (PFS), progressive disease (PD), and death (Fig.1). It is assumed that all patients enter the model in the PFS state. Each treatment cycle spans 3 weeks, and the simulation period for the entire study is set at 10 years. The primary outcomes of the model include total costs, quality-adjusted life years (QALYs), and the incremental cost-effectiveness ratio (ICER). In accordance with the "Chinese Guidelines for Pharmacoeconomic Evaluation 2020" [12], a discount rate of 5% was applied to both costs and utility values. A one-way sensitivity analysis was conducted within a range of 0–8%. The study adopts the perspective of China's healthcare system, with a time horizon of 10 years. The willingness-to-pay (WTP) threshold was established at three times China's per capita gross domestic product (GDP) in 2024, equating to 287,391 yuan/QALY.

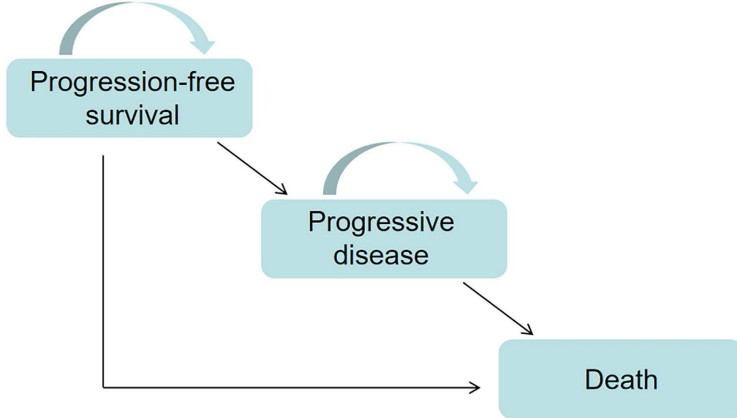

**Fig 1. The Markov model simulating outcomes for the NCT04878016 trial.**

## Survival Analysis

Data points were extracted from the overall survival (OS) and progression-free survival (PFS) curves reported in clinical trials [8] using WebPlotDigitizer version 4.7. Data reconstruction was performed using R 4.4.1 software, and five distribution parameter models, namely Exponential, Gompertz, Weibull, Log-logistic, and Lognormal, were used for fitting. The optimal fitting distribution was selected based on visual inspection of the fitting graphs and the Akaike information criterion (AIC) and Bayesian information criterion (BIC). The estimated graphs of OS and PFS curves for the two groups are shown in Fig.2, and the fitting distribution parameters of the survival curves are presented in Tables 1 and 2. In this study,

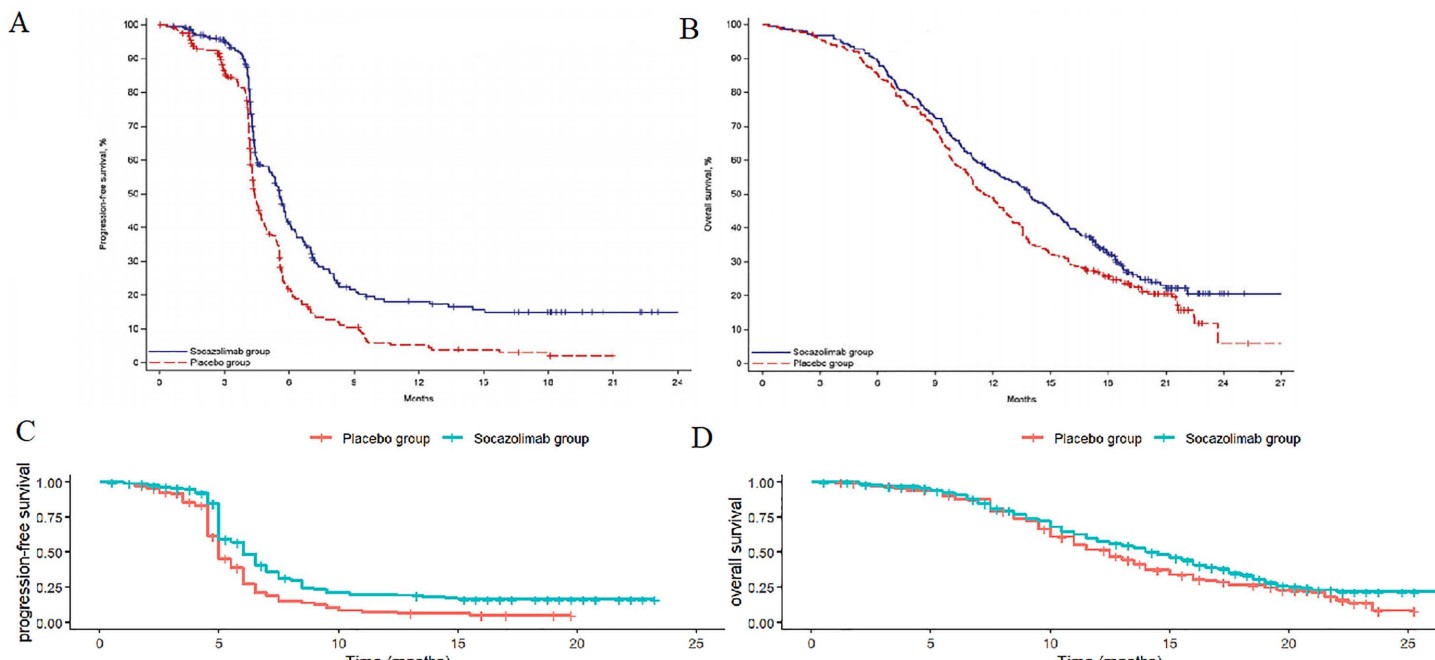

**Fig 2. Optimal curve fitting extrapolation of two treatment schemes. A.** Original PFS curve; **B.** Original OS curve; **C.** Simulated PFS curve; **D.** Simulated OS curve.

**Table 1. AIC and BIC of survival curve in two groups.**

| survival curve | Model criteria | Exponential | Gompertz | Weibull | Log-logistic | Lognormal |
|---|---|---|---|---|---|---|
| Treatment group | AIC | 843.465 | 838.231 | 797.499 | 733.288 | 744.971 |
| PFS curve | BIC | 846.979 | 845.258 | 804.526 | 740.315 | 751.998 |
| Control group | AIC | 819.229 | 782.680 | 718.929 | 659.669 | 681.126 |
| PFS curve | BIC | 822.742 | 789.707 | 725.956 | 666.696 | 688.153 |
| Treatment group | AIC | 1342.252 | 1282.369 | 1258.094 | 1251.844 | 1255.642 |
| OS curve | BIC | 1345.765 | 1289.396 | 1265.121 | 1258.870 | 1262.668 |
| Control group | AIC | 1394.877 | 1323.131 | 1294.772 | 1286.559 | 1297.517 |
| OS curve | BIC | 1398.391 | 1330.158 | 1301.799 | 1293.585 | 1304.544 |

AIC: Akaike information criterion, BIC: Bayesian information criterion, PFS: progression-free survival, OS: overall survival.

**Table 2. Parameter distribution of survival curve in two groups.**

| Survival curve | Optimal fitting distribution | Shape | Scale |
|---|---|---|---|
| Treatment group PFS curve | Log-logistic | 3.07673 | 6.88847 |
| Control group PFS curve | Log-logistic | 3.97539 | 5.39080 |
| Treatment group OS curve | Log-logistic | 2.63812 | 13.97048 |
| Control group OS curve | Log-logistic | 2.83146 | 12.47755 |

PFS: progression-free survival, OS: overall survival.

the Log-logistic distribution was ultimately selected to fit the PFS and OS curves of the sotagliflozin plus chemotherapy regimen and the placebo plus chemotherapy regimen.

## Cost and Utility Values

The study adopts the perspective of China's healthcare system, focusing solely on direct medical costs. Detailed cost items are provided in Table 3. Drug prices are based on the median of the winning bid prices for each province in 2024 as

**Table 3. Model parameters.**

| Variable | Baseline Value | Minimum | Maximum | Distribution | Reference |
|---|---|---|---|---|---|
| Cost (CNY) | | | | | |
| Socazolimab/mg | 115 | 92 | 138 | Gamma | https://www.yaozh.com/ |
| Etoposide/mg | 3.16 | 2.53 | 3.80 | Gamma | https://www.yaozh.com/ |
| Carboplatin/mg | 0.61 | 0.49 | 0.73 | Gamma | https://www.yaozh.com/ |
| Laboratory testing | 166 | 132.8 | 199.2 | Gamma | [20] |
| Radiological examinations | 300 | 240 | 360 | Gamma | [21] |
| Anemia | 1235.30 | 988.24 | 1482.36 | Gamma | [22] |
| Leukocytopenia | 2309.99 | 1847.99 | 2771.99 | Gamma | [23] |
| Decreased neutrophil count | 2877.40 | 2301.92 | 3452.88 | Gamma | [23] |
| Decreased platelet count | 822.89 | 658.31 | 987.47 | Gamma | [22] |
| BSC | 3115 | 2492 | 3738 | Gamma | [24] |
| Incidence rate of adverse reaction/% | | | | | |
| Decreased neutrophil count (Treatment group) | 69.1 | 55.28 | 82.92 | Beta | [8] |
| Leukocytopenia (Treatment group) | 44.2 | 35.36 | 53.04 | Beta | [8] |
| Anemia (Treatment group) | 23.3 | 18.64 | 27.96 | Beta | [8] |
| Decreased platelet count (Treatment group) | 33.7 | 26.96 | 40.44 | Beta | [8] |
| Decreased neutrophil count (Control group) | 66.8 | 53.44 | 80.16 | Beta | [8] |
| Leukocytopenia (Control group) | 34.0 | 27.2 | 40.8 | Beta | [8] |
| Anemia (Control group) | 20.6 | 16.48 | 24.72 | Beta | [8] |
| Decreased platelet count (Control group) | 27.9 | 16.48 | 24.72 | Beta | [8] |
| PFS | 0.673 | 0.538 | 0.808 | Beta | [16–17] |
| PD | 0.473 | 0.378 | 0.568 | Beta | [16–17] |
| Discount rate/% | 5 | 0 | 8 | Beta | [12] |
| Weight | 59 | 47.2 | 70.8 | Normal | [13] |
| Body surface area/m$^2$ | 1.72 | 1.38 | 2.06 | Normal | [14] |
| Creatinine clearance rate (ml/min) | 90 | 80 | 100 | Gamma | [15] |

BSC: best supportive treatment, PFS: progression-free survival, PD: progressive disease.

published by the https://www.yaozh.com/. The costs of the two treatment regimens were calculated according to the drug administration schedules used in the trial. For certain drugs, dosages were determined based on patient weight, body surface area (BSA), and creatinine clearance rate (CCR). In this study, it is assumed that the average patient weight is 59 kg [13], BSA is 1.72 m² [14], and CCR is 90 ml/min [15]. Since utility values were not reported in the original trial, the utility parameters used in this study were derived from published literature. Specifically, the utility value for the progression-free survival (PFS) state is 0.673, and for the progressive disease (PD) state, it is 0.473 [16–17]. The costs associated with adverse reactions were estimated by multiplying the incidence rates of adverse events by the cost of treating a single event. This study only includes adverse reactions of grade ≥ 3 with an incidence rate ≥ 5%.

## Sensitivity analysis

This study conducted both one-way sensitivity analysis and probabilistic sensitivity analysis to evaluate the robustness of the model. The one-way sensitivity analysis calculated the incremental cost-effectiveness ratio (ICER) using a ± 20% variation around the base values to assess the impact on cost-effectiveness when parameters such as drug costs, adverse reaction management costs, health utility values, and diagnostic testing costs vary within a specified range. The results of one-way sensitivity analysis were visualized using a tornado diagram. The probabilistic sensitivity analysis was performed through 1000 Monte Carlo simulations to sample the distributions of each parameter. Cost parameters were modeled using a Gamma distribution, while adverse reaction rates and utility parameters were modeled using a Beta distribution. The probabilistic sensitivity analysis results were presented in an incremental cost-effectiveness scatter plot and a cost-effectiveness acceptability curve.

# Results

## Base-case analysis results

The results of the base-case analysis showed that, compared with the chemotherapy-alone group, the combination of socazolimab and chemotherapy increased 0.21 QALYs, but the total cost also increased by 74,616.56 yuan. The ICER was 355,316.95 yuan/QALY, as shown in Table 4. The WTP set in this study was 287,391 yuan. Since the ICER was much higher than the WTP, the combination of socazolimab and chemotherapy was not economically viable.

## Price simulation

Through price simulation, we projected potential scenarios following the drug's inclusion in China's National Reimbursement Drug List negotiation. We established a series of hypothetical drug price discount rates and recalculated the ICER values as shown in Table 5. The results demonstrated that when the price of socazolimab decreased by 30%, 50%, and 70%, the corresponding ICER values were 293,555.90 yuan/QALY, 252,381.90 yuan/QALY, and 211,207.90 yuan/QALY, respectively. When the price of socazolimab was reduced by 50%, the ICER value fell below the WTP threshold

**Table 4. Baseline results.**

| Parameters | Treatment group | Control group |
|---|---|---|
| Total cost/CNY | 141709.54 | 67092.98 |
| Incremental cost/CNY | 74616.56 | |
| Effect/QALYs | 0.87 | 0.66 |
| Incremental effect/QALYs | 0.21 | |
| ICER, CNY/QALY | 355316.95 | |

CNY: Chinese yuan, QALY: quality-adjusted life year, ICER: incremental cost-effectiveness ratio.

**Table 5. ICER values at different discount rates for Socazolimab prices.**

| Price discount rate | Group | Total cost/CNY | Incremental cost/CNY | Effect/QALYs | Incremental effect/QALYs | ICER, CNY/QALY |
|---|---|---|---|---|---|---|
| 90% | Treatment group | 137386.27 | 70293.29 | 0.87 | 0.21 | 334729.95 |
| 80% | Treatment group | 133062.99 | 65970.01 | 0.87 | 0.21 | 314142.90 |
| 70% | Treatment group | 128739.72 | 61646.74 | 0.87 | 0.21 | 293555.90 |
| 68% | Treatment group | 127875.07 | 60782.09 | 0.87 | 0.21 | 289438.52 |
| 67% | Treatment group | 127442.74 | 60349.76 | 0.87 | 0.21 | 287379.81 |
| 60% | Treatment group | 124416.45 | 57323.47 | 0.87 | 0.21 | 272968.90 |
| | Control group | 67092.98 | | 0.66 | | |

CNY: Chinese yuan, QALY: quality-adjusted life year, ICER: incremental cost-effectiveness ratio.

of ¥268,074, indicating that the socazolimab plus chemotherapy regimen became cost-effective at this price point. Our analysis demonstrates that in order for the socazolimab plus chemotherapy regimen to be cost-effective at the 3×GDP threshold, its unit price would need to be reduced to approximately 77 yuan/mg.

## One-way sensitivity analysis

The tornado diagram from the one-way sensitivity analysis (Fig 3) indicates that the four factors with the greatest influence on the model results are the utility value in the PD state, the utility value in the PFS state, the cost of socazolimab, and the cost of managing neutropenia. Other variables have a relatively minor impact on the ICER value; changes within their respective ranges do not cause the ICER to exceed the WTP threshold. When parameters vary within the specified

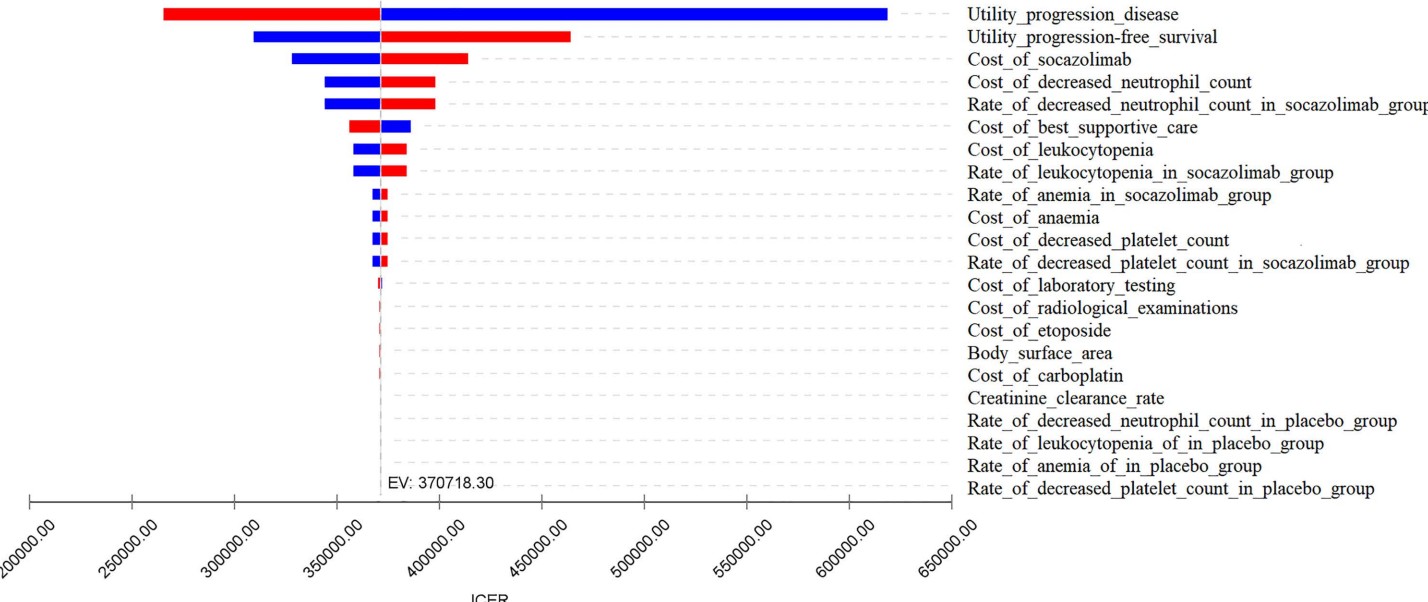

**Fig 3. Tornado diagram for one-way sensitivity analysis.**

ranges, the ICER remains significantly higher than the WTP, indicating a lack of economic feasibility. These findings are consistent with the conclusions drawn from the base-case analysis.

## Probabilistic sensitivity analysis

The cost-effectiveness acceptability curve (Fig 4) demonstrates that the economic efficiency of the socazolimab plus chemotherapy regimen for ES-SCLC increases as the willingness-to-pay (WTP) threshold rises, while the chemotherapy-alone group exhibits an opposite trend. Specifically, when the WTP threshold is approximately 150,000 yuan, the probability of the socazolimab plus chemotherapy regimen being cost-effective is 0%. When the WTP threshold reaches around 397,500 yuan, the probabilities of both regimens being cost-effective are equal. These findings from the probabilistic sensitivity analysis are consistent with the results of the base-case analysis. Additionally, we calculated the cost-effectiveness probability for the socazolimab group, which was 0% at the 1 × GDP threshold (95,797 CNY/QALY) and 2.8% at the 2 × GDP threshold (191,594 CNY/QALY).

The incremental cost-effectiveness scatter plot (Fig 5) shows that the majority of the ICER values generated from 1,000 Monte Carlo simulations fall above the WTP line, indicating that the socazolimab plus chemotherapy regimen does not offer a cost-effective advantage.

## Discussion

The importance of pharmacoeconomic evidence in supporting decision-making for the efficient allocation of national healthcare resources is increasingly recognized, particularly in the evaluation of new drugs that offer significant clinical benefits but come with higher costs. Additionally, pharmacoeconomic analysis facilitates long-term planning and policy development in public health, medical insurance, and related fields.

This study, based on the NCT04878016 research, takes the perspective of China's healthcare system and uses a partitioned survival model to evaluate the cost-effectiveness of socazolimab combined with chemotherapy as a first-line treatment for ES-SCLC. The results of the one-way sensitivity analysis show that the parameters that have a significant

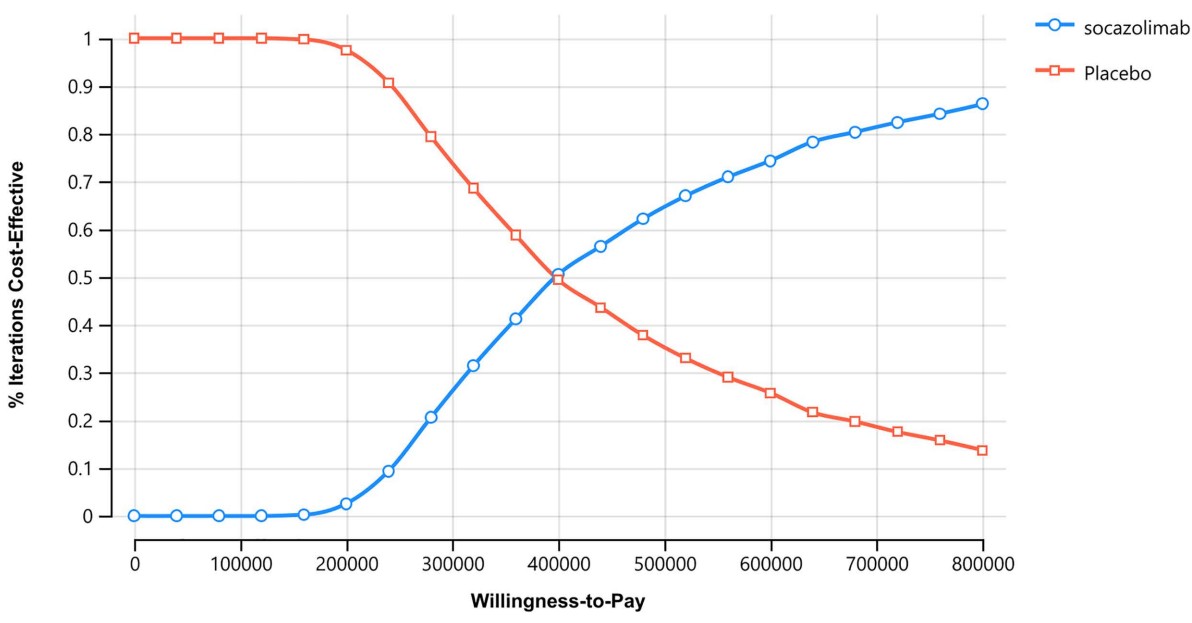

**Fig 4. Acceptability curves.**

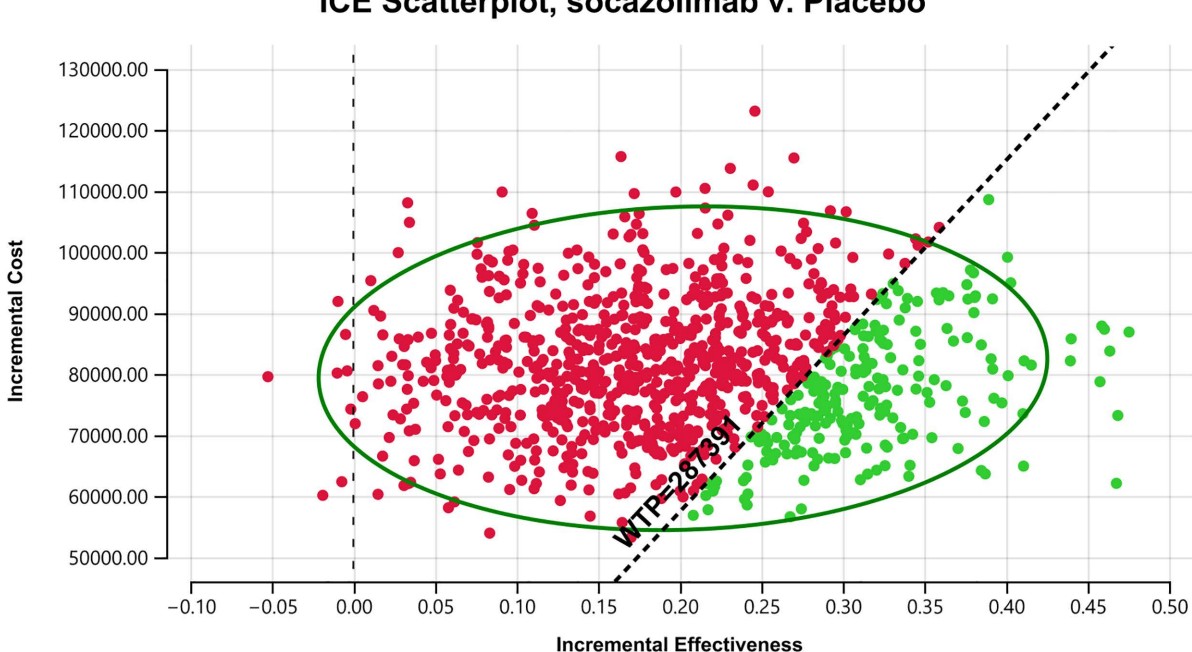

**Fig 5. Cost-effective scatter plot.** Results of Monte Carlo probabilistic sensitivity analysis showing incremental cost-effectiveness of socazo-limab + chemotherapy versus placebo + chemotherapy.

impact on the incremental cost-effectiveness ratio (ICER) value are the utility values of the progressive disease (PD) state and the progression-free survival (PFS) state, the cost of socazolimab, and the cost of neutropenia management. These findings underscore the importance of not only extending patient survival but also considering patient quality of life and the economic burden of subsequent anticancer treatments. The results of the probabilistic sensitivity analysis show that the majority of the ICER values generated by 1,000 Monte Carlo simulations are above the willingness-to-pay (WTP) line (three times the per capita GDP of China in 2024), indicating that the combination of socazolimab and chemotherapy is not economically advantageous. When the willingness-to-pay (WTP) threshold is approximately 397,500 yuan, the probability of economic effectiveness is equal between the combination therapy of socazolimab and chemotherapy and chemotherapy alone. Threshold analysis reveals that socazolimab would require a price reduction from its current level to achieve cost-effectiveness at China's willingness-to-pay threshold of 3 × GDP per capita. This finding can provide import-ant quantitative reference for future National Reimbursement Drug List negotiations.

When the willingness-to-pay value is greater than 397,500 yuan, the combination of socazolimab and chemotherapy is economically effective. However, in China, the economic development among different provinces is highly uneven. For Beijing, Shanghai, Jiangsu, Fujian, Zhejiang, and Tianjin, the willingness-to-pay value is all greater than 397,500 yuan [18], and the economic effectiveness results will be reversed.

Although a cost-effectiveness analysis [19] based on the same Phase III clinical trial (NCT04878016) has been pub-lished, our study employs a distinct modeling approach—the partitioned survival model (PSM). The PSM utilizes recon-structed individual patient data from Kaplan-Meier curves to directly simulate the proportion of patients in different health states over time, without requiring the assumption of constant transition probabilities between states. This approach offers greater flexibility in capturing the complex relationship between progression-free survival and overall survival when evaluating immuno-oncology therapies. In contrast, the Markov model used in the published study relies on estimating

monthly transition probabilities between health states. While the core clinical findings are consistent, the use of different model structures for cross-validation strengthens the robustness and reliability of the cost-effectiveness analysis results. In addition, a key novel aspect of our analysis is the exploration of variable WTP thresholds reflective of China's significant regional economic disparities. Unlike a one-size-fits-all approach, our findings provide tailored insights for policymakers in more affluent provinces.

This study has several limitations. Firstly, the progression-free survival (PFS) and overall survival (OS) curves were extrapolated using parameter distribution fitting. While this approach allows for projecting survival trends beyond the trial period of the NCT04878016 study, it also introduces additional uncertainty into the model results. Secondly, this study only considered adverse events of grade ≥ 3 with an incidence rate ≥ 5%. However, sensitivity analysis revealed that variations in the probability of adverse events did not significantly impact the overall model outcomes. Besides, our cost analysis was confined to direct medical costs and did not include indirect costs (such as productivity losses and caregiver burden). From a societal perspective, our study may underestimate the true economic burden of extensive-stage small cell lung cancer. Future research should aim to adopt a societal perspective by prospectively collecting data on productivity losses and caregiver burden to provide a more comprehensive economic evaluation. The last one, the exclusion of low-grade chronic immune-related adverse events (irAEs) from the model represents a limitation, as this may lead to an underestimation of the total treatment-related burden. However, it should be emphasized that this approach makes the conclusions of our base-case analysis more conservative and robust. Future research, upon obtaining more detailed patient-reported outcome data and long-term follow-up data, could conduct more refined analyses on this aspect.

## Conclusion

Socazolimab combined with chemotherapy is not economically advantageous compared to chemotherapy alone as a first-line treatment for ES-SCLC from the perspective of the Chinese healthcare system. However, due to the uneven economic development across different regions in China, this plan will be economically viable in regions where WTP exceeds 397,500 yuan.

## Author contributions

**Conceptualization:** Panfeng Feng.

**Data curation:** Lian Tang, Yong Chen, Shaoqing Zhan, Longxun Zhu, Panfeng Feng.

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
