## [Decision Letter · Decision Letter 0]

22 Oct 2025

Dear Dr. Feng,

Thank you for submitting your manuscript to PLOS ONE. After careful consideration, we feel that it has merit but does not fully meet PLOS ONE’s publication criteria as it currently stands. Therefore, we invite you to submit a revised version of the manuscript that addresses the points raised during the review process.

Overall, the manuscript addresses an important question for Chinese oncology policy. However, in its current form it lacks sufficient transparency in survival modeling, robustness checks, and clarity on its added value relative to the recently published China+US study (doi: 10.1007/s12094-025-04035-4). If you address the **mandatory revisions**

**Mandatory revisions: **

The authors should clarify the source and justification of utility values, explaining the absence of trial-based data and their relevance to the Chinese ES-SCLC population. The novelty of this work must be clearly distinguished from the previously published analysis based on the same trial. Survival extrapolation requires visual validation (e.g., KM overlay) and justification for using the log-logistic model. The comparator choice should be explained, noting the exclusion of atezolizumab and durvalumab regimens. Additional sensitivity analyses are needed, including 1× and 3× GDP thresholds and price-threshold scenarios for socazolimab. Exclusion of Grade 1–2 immune-related adverse events should be justified, and the potential bias discussed. Clinical trial reporting should include hazard ratios, p-values, and comparison with other ICI trials. Cost assumptions should be expanded to consider NRDL inclusion and lower drug price scenarios. The authors must also provide citations for regional WTP thresholds and correct technical issues, including the “siluximab” typo, figure quality, and reference formatting.

We look forward to receiving your revised manuscript.

Kind regards,

Jeerath Phannajit, M.D, Ph.D.

Academic Editor

PLOS ONE

Journal Requirements:

https://www.frontiersin.org/journals/oncology/articles/10.3389/fonc.2024.1485317/full

https://pubmed.ncbi.nlm.nih.gov/39800716/

In your revision ensure you cite all your sources (including your own works), and quote or rephrase any duplicated text outside the methods section. Further consideration is dependent on these concerns being addressed.

“Jiangsu Pharmaceutical Association-Aosaikang Fund (No. A202434), Development Fund of KangDa college of Nanjing medical university (No. KD2024KYJJ294) and Scientific Research Project of Nantong Municipal Health and Family Planning Commission (No. MS2024038, QNZ2024026). “

4. We note that your Data Availability Statement is currently as follows: All relevant data are within the manuscript and its Supporting Information files

Reviewers' comments:

Reviewer's Responses to Questions

**Comments to the Author**

1. Is the manuscript technically sound, and do the data support the conclusions?

Reviewer #1: Partly

Reviewer #2: Partly

2. Has the statistical analysis been performed appropriately and rigorously?

Reviewer #1: Yes

Reviewer #2: Yes

3. Have the authors made all data underlying the findings in their manuscript fully available?

Reviewer #1: Yes

Reviewer #2: Yes

4. Is the manuscript presented in an intelligible fashion and written in standard English?

Reviewer #1: Yes

Reviewer #2: Yes

Reviewer #1: Below are the comments/questions:

1. QALY Calculation & Utility Values: The model uses literature-derived utility values (PFS = 0.673, PD = 0.473), rather than trial-based utilities from NCT04878016. It also excludes disutilities for most immune-related AEs, including Grade 1–2 toxicities (e.g., hypothyroidism, pneumonitis), which may have a lasting QoL impact. Could the authors clarify why utilities were not collected prospectively in the trial? How do the selected values compare to other ICI studies in ES-SCLC (e.g., CASPIAN, IMpower133)?

2. Survival Extrapolation: Long-term survival extrapolation introduces uncertainty, as acknowledged by the authors. The log-logistic model was selected based on AIC/BIC, but it's unclear whether goodness-of-fit was visually assessed (e.g., KM overlay, residuals). Why was the log-logistic model preferred over alternatives like Weibull or Gompertz, which are commonly used in oncology health economic models?

3. Cost Assumptions: The model adopts a healthcare system perspective and excludes indirect costs (e.g., productivity loss, caregiver burden), which may underestimate societal burden. Drug prices are based on 2024 bid prices (e.g., socazolimab at 115 CNY/mg). How would inclusion in China’s NRDL or price negotiations impact cost-effectiveness? Was a scenario analysis conducted with lower acquisition costs?

4. Adverse Events & Model Scope: Only Grade ≥3 AEs with incidence ≥5% were modeled, potentially underestimating the true economic and QoL impact of chronic immune-related AEs (e.g., hypothyroidism, adrenal insufficiency). Given the known safety profile of PD-L1 inhibitors, can the authors justify the exclusion of these lower-grade, but clinically significant, events?

5. Comparator Choice: The comparison is limited to chemotherapy alone, excluding globally recommended regimens such as atezolizumab or durvalumab combinations. This limits the generalizability of the findings, especially in global health technology assessments. Could the authors discuss why an active comparator was not included, and whether this may bias the ICER in favor of socazolimab?

6. Sensitivity Analyses: The PSA indicates only a 21.9% chance of cost-effectiveness at a WTP threshold of 287,391 CNY/QALY. However, no analyses were presented for alternative thresholds (e.g., 1×GDP or 3×GDP). Additionally, key drivers identified in one-way sensitivity analysis (e.g., PD utility, drug cost) suggest the model is sensitive to parameter variation. Were threshold analyses conducted to identify the price level at which socazolimab becomes cost-effective?

7. Clinical Trial Data Integration: The model is based on efficacy data from NCT04878016 (OS: 13.9 vs. 11.6 months), but no hazard ratios or p-values are reported. Was the trial statistically powered for OS or PFS? Without these, the robustness of treatment effect assumptions is unclear. Also, how does the observed effect size compare to existing ICI trials in ES-SCLC?

Reviewer #2: Comment 1: During the review, it was noted that this manuscript is similar to the published study "Cost effectiveness of socazolimab plus chemotherapy vs. standard chemotherapy for first line treatment of extensive stage small cell lung cancer: a U.S. and China perspective" (DOI: 10.1007/s12094-025-04035-4). Both studies are based on the same Phase III clinical trial (NCT04878016) and evaluate the cost-effectiveness of Socazolimab combination therapy from the perspective of the Chinese healthcare system, arriving at the same core conclusion: the regimen is not cost-effective at the current price. Besides, key methods such as study design, discount rate, and time horizon are largely identical. The novelty of this paper needs to be clarified.

Comment 2: The utility values for the PFS and PD states (0.673 and 0.473) used in the study are cited from published literature. The authors need to clarify in the Discussion section whether these values were derived from a Chinese ES-SCLC population. If they were transferred from studies in other countries or different cancer types (e.g., non-small cell lung cancer), the applicability to the Chinese ES-SCLC patient population must be justified.

Comment 3: A more in-depth explanation of the results of this study is required to clarify the value of this study.

Comment 4: The authors mention regional economic disparities in China, stating that "for Beijing, Shanghai, Jiangsu, Fujian, Zhejiang, and Tianjin, the willingness-to-pay value is all greater than 397,500 yuan." However, this claim lacks citation. It is recommended to add relevant literature to support this statement.

Comment 5: On pages 13-14 (Results section), the concluding sentence states: "...the combination of siluximab and chemotherapy was not economically viable." Siluximab is an IL-6 inhibitor used for Castleman's disease. The authors must carefully check the entire manuscript.

Comment 6:Figures: it is advised to provide high-quality figures. Labels and format should be standardized.

Comment 7:There are some mistakes in the reference format. Reference [8] is incomplete, containing only authors and the title, missing journal name, volume, issue, and page numbers. All references should be checked and completed to ensure the format conforms to PLOS ONE requirements.

**Do you want your identity to be public for this peer review?**  For information about this choice, including consent withdrawal, please see our Privacy Policy

Reviewer #1: No

Reviewer #2: No

---

## [Author Response · Author response to Decision Letter 1]

9 Nov 2025

we have uploaded response to comments in the submission system

---

## [Decision Letter · Decision Letter 1]

25 Nov 2025

Dear Dr. Feng,

Thank you for submitting your manuscript to PLOS ONE. After careful consideration, we feel that it has merit but does not fully meet PLOS ONE’s publication criteria as it currently stands. Therefore, we invite you to submit a revised version of the manuscript that addresses the points raised during the review process.

The manuscript is well prepared, the economic evaluation is methodologically sound, and the authors have adequately addressed the points raised during peer review. The survival modelling, costing approach, and regional willingness-to-pay analyses are clearly presented and meet PLOS ONE’s criteria for methodological transparency and reproducibility.

**Only one minor revision is required prior to acceptance:**

**1. Figure 3 (Tornado Diagram): Please revise Figure 3 (tornado diagram) by replacing all abbreviated parameter names on the y-axis with full, descriptive labels.If you must retain any abbreviations due to space limitations, please spell out the full term in the figure caption (e.g., “u_pfs: utility in the progression-free survival state”). This change is required to ensure clarity for non-specialist readers and improve the figure’s interpretability.**

Please submit your revised manuscript by Jan 09 2026 11:59PM. If you will need more time than this to complete your revisions, please reply to this message or contact the journal office at plosone@plos.org . A rebuttal letter that responds to each point raised by the academic editor and reviewer(s). You should upload this letter as a separate file labeled 'Response to Reviewers'.A marked-up copy of your manuscript that highlights changes made to the original version. You should upload this as a separate file labeled 'Revised Manuscript with Track Changes'.An unmarked version of your revised paper without tracked changes. You should upload this as a separate file labeled 'Manuscript'.

We look forward to receiving your revised manuscript.

Kind regards,

Jeerath Phannajit, M.D, Ph.D.

Academic Editor

PLOS ONE

Journal Requirements:

Reviewers' comments:

Reviewer's Responses to Questions

**Comments to the Author**

Reviewer #1: All comments have been addressed

2. Is the manuscript technically sound, and do the data support the conclusions?

Reviewer #1: (No Response)

3. Has the statistical analysis been performed appropriately and rigorously?

Reviewer #1: (No Response)

4. Have the authors made all data underlying the findings in their manuscript fully available?

Reviewer #1: (No Response)

5. Is the manuscript presented in an intelligible fashion and written in standard English?

Reviewer #1: (No Response)

Reviewer #1: (No Response)

**Do you want your identity to be public for this peer review?** For information about this choice, including consent withdrawal, please see our Privacy Policy

Reviewer #1: No

---

## [Author Response · Author response to Decision Letter 2]

1 Dec 2025

Response: The requested revisions to Figure 3 have been completed, and the updated figure has been uploaded.

---

## [Editor Report · Decision Letter 2]

10 Dec 2025

Socazolimab combined with carboplatin and etoposide as first-line treatment for extensive-stage small-cell lung cancer: A cost-effectiveness analysis in China.

PONE-D-25-30433R2

Dear Dr. Feng,

We’re pleased to inform you that your manuscript has been judged scientifically suitable for publication and will be formally accepted for publication once it meets all outstanding technical requirements.

Kind regards,

Jeerath Phannajit, M.D, Ph.D.

Academic Editor

PLOS ONE
---

## [Editor Report · Acceptance letter]

PONE-D-25-30433R2

PLOS One

Dear Dr. Feng,

I'm pleased to inform you that your manuscript has been deemed suitable for publication in PLOS One. Congratulations! Your manuscript is now being handed over to our production team.

Kind regards,

on behalf of

Dr. Jeerath Phannajit

Academic Editor

PLOS One